# Class-Aware Fish Species Recognition Using Deep Learning for an Imbalanced Dataset

**DOI:** 10.3390/s22218268

**Published:** 2022-10-28

**Authors:** Simegnew Yihunie Alaba, M M Nabi, Chiranjibi Shah, Jack Prior, Matthew D. Campbell, Farron Wallace, John E. Ball, Robert Moorhead

**Affiliations:** 1Department of Electrical and Computer Engineering, James Worth Bagley College of Engineering, Mississippi State University, Starkville, MS 39762, USA; 2Northern Gulf Institute, Mississippi State University, Starkville, MS 39759, USA; 3NOAA—National Marine Fisheries Service, Southeast Fisheries Science Center, 3209 Frederic Street, Pascagoula, MS 39567, USA; 4NOAA Fisheries, 4700 Avenue U, Galveston, TX 77551, USA

**Keywords:** class-aware loss, deep learning, fish recognition, imbalanced data, object detection, species classification

## Abstract

Fish species recognition is crucial to identifying the abundance of fish species in a specific area, controlling production management, and monitoring the ecosystem, especially identifying the endangered species, which makes accurate fish species recognition essential. In this work, the fish species recognition problem is formulated as an object detection model to handle multiple fish in a single image, which is challenging to classify using a simple classification network. The proposed model consists of MobileNetv3-large and VGG16 backbone networks and an SSD detection head. Moreover, a class-aware loss function is proposed to solve the class imbalance problem of our dataset. The class-aware loss takes the number of instances in each species into account and gives more weight to those species with a smaller number of instances. This loss function can be applied to any classification or object detection task with an imbalanced dataset. The experimental result on the large-scale reef fish dataset, SEAMAPD21, shows that the class-aware loss improves the model over the original loss by up to 79.7%. The experimental result on the Pascal VOC dataset also shows the model outperforms the original SSD object detection model.

## 1. Introduction

Fish species classification is an essential component of fisheries management and environmental monitoring. Accurate, reliable, and efficient species recognition of fish is necessary to identify the endangered species, determine the optimal harvest size or time, monitor ecosystems, and develop an intelligent production management system [1,2]. Because of the legal constraints on fishing techniques, precise fish species recognition is essential, especially when their survival is endangered or threatened. Most fisheries use the traditional form of species identification, which demands intensive human labor, consumes time, and can affect the fish’s normal behavior. The conventional approaches challenge fishery observers to maintain high-quality data for managing sustainability in the fishery industry, monitoring federal fisheries, assessing fish populations, and identifying different fish species. However, robust deep learning (DL) based fish species identification models would decrease cost and time and improve identification precision.

A machine vision solution can be implemented to replace the manual system. For fish detection, several approaches have been developed, including lidar [3,4], sonar [5], and RGB imaging [6]. RGB imagery is the preferred option to identify fish based on texture, color, and geometry due to ease of operation, cheaper cost, a lightweight system, and no destruction of fish habitat. In the computer vision community, different camera systems have been implemented to monitor the abundance of fish to aid in assessing the stock and sustainability of marine ecology [7,8,9,10]. Different DL techniques, such as object detection and classification, can be implemented to gather information regarding marine life [11,12]. However, the underwater environment has low light conditions, noise, and low-resolution images and videos, making it difficult to distinguish the fish from the background. Additionally, the movement of fish in the water results in images of fish in different shapes and introduces occlusion issues. These issues are challenging for underwater fish species identification and detection.

DL has significantly grown in computer vision applications to solve detection, localization, estimation, and classification problems [13,14,15,16]. However, the role is limited in marine ecology and agriculture [17,18] related applications. Different machine learning (ML) and DL methods have been proposed for fish species classification. Huang et al. [19] proposed the hierarchical features of fish and support vector machine (SVM) for fish classification on the fish4knowledge [20] dataset. Jager et al. [21] used AlexNet architecture for feature extraction and multiclass SVM for classification of the LifeCLEF 2015 [22] fish dataset. Zhuang et al. [23] utilized pre-processing and post-processing on modern DL-based models on LifeCLEF 2015 fish data. Most existing DL methods use a simple classification dataset with one fish in each image, which is not always the case in a real environment. In the real system, multiple fish in a single image results in the difficulty of applying a simple classification network. Our dataset, the Southeast Area Monitoring and Assessment Program Dataset (SEAMAPD21) [7], contains multiple fish in a single image, representing a natural habitat. We formulate the fish species identification task as an object detection problem to solve the above problems. Two feature extraction networks, MobileNetv3-large [24] and VGG16 [25], are used as feature extraction networks. The mobileNetv3-large network has fewer parameters than the VGG16 network but has lower mean average precision (mAP), which shows the performance and network speed trade-off. The single-shot multibox detector (SSD) detection head is also used as a regression and classification network. The network generates the class confidence of each species and the location in the image (bounding box information).

### Contributions

We formulate the fish species recognition problem as an object detection problem to handle multiple fish recognition in a single image in real time. Additionally, a new loss function is proposed to handle the class imbalance problem to avoid the model’s bias toward the dominant class. The proposed model architecture is shown in Figure 1. Our main contributions can be summarized as follows.
We formulate the classification problem as an object detection problem to handle multi-class classification and multiple fish in a single image. The model not only classifies the fish species but also localizes each fish in the image, which can work in videos as well.The designed class-balanced term helps to build a class-aware loss function to handle the class imbalance problem. The significance of the loss function is shown in the experiment in Section 5.The model was trained with different backbone networks to show the trade-off between performance and speed between networks. If our need is performance only, we can pick the model with high performance. However, we can choose the lightweight model if we aim to design a fast model.The model was trained on the SEAMAPD21 dataset. The experimental results show the model’s promising performance on the challenging dataset.

The rest of the paper is organized as follows. Section 2 presents related work. The use of DL in the fishery industry is summarized in Section 3. Section 4 summarizes the proposed model architecture and sub-networks, such as feature extraction and detection head, dataset augmentation, and loss functions. Results and analyses of the experiment are presented in Section 5. Section 6 concludes the work.

## 2. Related Work

Fish classification is an extensively studied problem in image segmentation, pattern recognition, and information retrieval. Fish classification uses the resemblance with the representative specimen image to identify and categorize the target fish into species [26]. Some image processing-based techniques for fish recognition have been developed. A balance-guaranteed optimized tree [19] algorithm was developed to reduce the accumulation error in the detection process. Color and shape descriptors were employed to distinguish fish in RGB imagery [27,28]. In past studies, the features used to identify fish primarily relied on hand-crafted feature-generation techniques. If the essential features are missed, the accuracy of fish detection drops drastically. Furthermore, these shallow learning methods do not scale with data. Because of the deep layer structure and massive data support, the performance of DL approaches is higher than the shallow learning methods [29,30,31].

Instead of the manually created features generally used in classical ML techniques, DL-based vision algorithms distinguish objects by implicitly extracting their distinctive features. Nery et al. [32] presented a fish classification methodology based on a robust feature selection technique. The authors proposed a general set of features and their corresponding weights that the classifier used as prior information to analyze their impacts on the whole classification task. Villon et al. [6] compared DL and histogram of oriented gradients (HOG) and support vector machine (SVM) methods for coral reef fish detection and recognition in underwater videos. For the first method, DL extracts features for detection and classification. For the second method, features are extracted using HOG and fed into SVM for classification. Some studies, such as Li et al. [33], considered fish species classification as object detection. The authors presented fish detection and twelve species recognition of underwater images using Faster-RCNN [34]. However, the model is slow because Faster-RCNN is a two-stage object detection network. Two-stage object detection networks perform better than single-stage networks [35] such as you only look once (YOLO) [36] and SSD [37]. However, they are slower due to extensive computation in the two-stage process region proposal, classification, and regression networks [35,38].

In a real-world application, an unbalanced class distribution in a given dataset, sometimes called a long-tailed data distribution, where a few classes account for most of the data while most classes are under-represented, is common. Models that are trained on such datasets are biased toward the dominant classes. Most existing solutions adopt class rebalancing strategies such as resampling and reweighting based on the number of observations for each class. Sampling techniques, such as synthetic minority over-sampling technique (SMOTE) [39] samples by interpolating from neighboring samples or synthesizing for minor classes [40,41]. These solutions may not work for all datasets with class imbalance problems and may cause models to overfit. Cui et al. [42] proposed a class-balanced loss to solve the class imbalance problem. The authors trained the model on the CIFAR-10 and CIFAR-100 datasets [43] and showed a performance improvement over the original implementation. Li et al. [44] also proposed a balanced group softmax to solve long-tail distribution problems through group-wise training in object detection. This technique divides classes into disjoint groups, and the softmax operation is performed separately on each class. Hence, only classes with a similar number of training samples compete within each group. In this work, we propose class-aware loss to solve the class imbalance problem based on the inverse of the number of samples in each class. We also extend the class-balanced loss into the object detection problem to reweight the classification loss and the localization loss (the class-balanced loss was proposed for the classification problem). Therefore, reweighting the loss based on the inverse number of samples in each class minimizes the effect of model biases toward the dominant class.

## 3. Deep Learning for Fishery: Motivation

Accurate fish species recognition can provide data for identifying the abundance of fish species in a specific area, assisting with data used for controlling production management and ecosystem monitoring, especially identifying endangered species. Building a deep learning-based detection system helps maintain high-quality data for sustainable management in the fish industry. This technology reduces time, human labor, and cost to collect and analyze data. Additionally, the deep learning system does not affect the normal behavior of the fish in the habitat during data collection or monitoring. The traditional form of fishery management, including fish species identification, is labor intensive, expensive, and affects the regular activity of the fish in the habitat. Correct data for each species help determine the optimal harvest size and fishing time based on the species distribution in the area. Overall, it is essential to monitor the ecosystem.

Fish species can be distinguished by their shape, color, and size. However, similarity in shape and pattern of some species, relatively low resolution and contrast, and the change in light intensity and fish motion makes accurate fish species identification challenging [45,46]. These challenges complicate solving the problem using classical ML classification techniques (using hand-extracted features). DL methods can learn the unique features without hand-engineered features. Another problem in fish species identification is the number of fish species in each image. In most datasets, each image comprises a single fish, making the classification problem convenient, but finding a single fish in an image with multiple fish is not easy. The dataset of the SEAMAPDP21 [7] consists of many fish species in a single image, making it difficult to use a simple classification network. Therefore, we formulated the fish species identification problem as an object detection problem.

We design an object detection network for fish species recognition based on the SSD [37] object detection network. SSD, which consists of VGG16 [25] as a backbone and a detection head, is designed to do general object detection. In object detection, accuracy and speed are essential. Therefore, the VGG16 [25] backbone network is replaced by MobileNetv3-large [24] to make the model lightweight. The modified model consists of mobileNetv3 as a backbone network and SSD detection head as a classification and regression network, as shown in Figure 1.

## 4. The Proposed Method and Implementation Details

The proposed model consists of a feature extraction network and a detection head, as shown in Figure 1.

**Figure 1 sensors-22-08268-f001:**
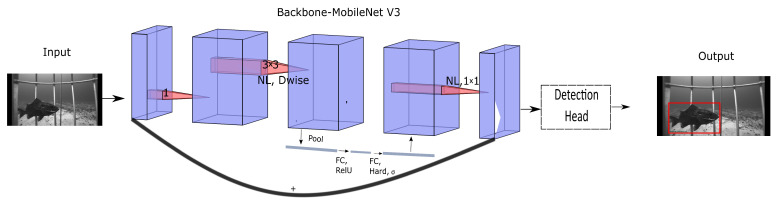
Proposed architecture. The architecture comprises the MobileNetv3-large [24] backbone and SSD [37] detection head. The MobileNetv3 feature extraction network is a lightweight model with good feature extraction capability. The extracted high-level features are input to the SSD detection head for classification and regression tasks. The network outputs fish species type and bounding box information for each image. The model is also trained with the VGG16 [25] backbone network.

The MobileNetv3-large [24] is used as a feature extraction network to extract important features from images and feed them into the detection head for species classification and regression. Once the features are extracted, the SSD [37] detection head uses the high-level feature to predict species classes and regresses bounding boxes for each image. Four boxes with different aspect ratios and scales for each location are predicted, as shown in Figure 2. Each default box has a predicted confidence score and localization offsets for all object categories.

### 4.1. Feature Extraction Network

Due to the rapid growth of DL, many feature extraction networks, such as VGG [25], ResNet [47], EfficinetNet [48], and MobileNet [49], have been developed. We use MobileNetv3-large, a variety of MobileNet [49], as a feature extraction network because of its lightweight parameter size and good feature extraction capability. The number of parameters of VGG16 [25] is more than 27 times the amount for MobileNetv3-large [24]. The MobileNetv3 network searches the global network structures using platform-aware network architecture search (NAS) by optimizing the network block-wise. Next, the NetAdapt algorithm [50] is used to search the number of filters per layer. This method is essential to fine-tune each layer sequentially, complementing platform-aware NAS. The NAS and NetAdapt techniques can be combined to find an optimized model for specific hardware. The hard-swish [51] activation function used in the MobileNetv3 network also has different advantages, especially during the deployment phase, such as minimizing a potential numerical precision loss usually caused by the sigmoid function. The network also uses depth-wise separable convolution instead of the standard convolution. The depth-wise separable convolution consists of the depth-wise convolution to filter the input channels and point-wise convolution to combine the output of depth-wise convolution. Overall, the depth-wise separable convolution has less computational cost than the standard convolution. These features of the MobileNetv3 network are essential for a real-time deployment of the MobileNetv3-based models on low-power devices. The model is also trained using VGG16 [25] as a backbone network.

### 4.2. Detection Head

The SSD [37] detection head is used for species prediction and bounding box regression. The SSD detection head makes four object predictions for each location at different scales and aspect ratios [37,52]. This property is essential for detecting small and large objects, which makes SSD preferable as a detection head [52]. A non-maximal suppression (NMS) is also used to suppress redundant predictions based on an intersection over union (IOU) value. The IOU evaluation metric measures the overlap between the ground truth and prediction regions. A higher IOU value indicates a more correct prediction. In most object detection networks, an IOU of 0.5 is commonly used. We also use an IOU of 0.5 to measure a true positive prediction.

### 4.3. Dataset and Data Augmentation

The large-scale reef fish SEAMAPD21 [7] dataset used for the experiment includes 130 species and 28,328 images; however, it is an imbalanced dataset with some species having many samples, whereas others have few, as shown in Figure 3.

The dataset is collected from the Gulf of Mexico continental shelf from Brownsville, TX, USA, to the Dry Tortugas, FL, USA. The target area is also widely diverse in water depth, ranging from 15 m to 200 m [7]. All the sample images are RGB images. When there is a class imbalance, the network prediction is influenced by species with many samples. We modified the cross-entropy loss to handle the class imbalance problem (see Section 4.4). The dataset is fishery-independent (a fishery-independent dataset is a dataset recorded from fish habitat, underwater), so it is challenging to detect fish because of the low resolution and similarity between the background and images (see Ref. [7] for details about the dataset). Sample images from the dataset are shown in Figure 4. Detecting the fish in some images is challenging, even for a human. The dataset is divided into training, validation, and testing in a 70/15/15 ratio.

Data augmentation, which is well-studied in classification problems, is used to increase the dataset slightly using various techniques, such as color transformation, random rotation, rescaling, cropping, zooming, and contrast changes [52]. Different augmentation techniques are available for classification and object detection. We apply random cropping, horizontal flipping with a probability of 0.5, and photometric distortion, including random brightness change, random contrast, and random hue, in the training dataset.

### 4.4. Loss

The loss function determines the difference between the network-predicted value and the ground truth. If the loss value is large, the predicted value differs from the ground truth, i.e., the prediction is not correct. Therefore, the network trains to learn and updates parameters (weights and biases) to minimize the loss function. Focal loss [53] is a commonly used classification loss for class imbalance, mainly foreground–background class imbalance. However, it does not give better accuracy than cross-entropy in our model because our class imbalance is foreground–foreground class imbalance. Therefore, we use cross-entropy loss as a classification loss and smoothL1 loss [54] as a localization loss. The localization loss, which is similar to SSD [37], can be defined as:(1)$$\begin{array}{cc} L_{loc(x,p,g)} & =\sum_{i\epsilon pos}^{N}\sum_{m\epsilon (c_{x},c_{y},w,h)}x_{ij}^{k}SmoothL1\left(p_{i}^{m}-{\hat{g}}_{j}^{m}\right)\\ {\hat{g}}_{j}^{c_{x}} & =\left(g_{j}^{c_{x}}-d_{i}^{c_{x}}\right),\\ {\hat{g}}_{j}^{c_{y}} & =\left(g_{j}^{c_{y}}-d_{i}^{c_{y}}\right),\\ {\hat{g}}_{j}^{w} & =\log\left(\frac{g_{j}^{w}}{d_{i}^{w}}\right),\\ {\hat{g}}_{j}^{h} & =\log\left(\frac{g_{j}^{h}}{d_{i}^{h}}\right), \end{array}$$
where *N* is the number of matched default boxes, *d* is the default box/anchor, *p* is the predicted box, *g* is the ground truth box, *w* is width, and *h* is height. The expression $x_{ij}^{k}$ matches the *i*th default box to the *j*th ground truth box of category *k*, and $c_{x}$ and $c_{y}$ are centers of the bounding box. Similarly, the classification loss is:(2)$$L_{cls}=-\frac{1}{n}\left(\sum_{i=1}^{N}y_{i}log\left(\hat{y_{i}}\right)\right),$$
where *n* is total training samples, $y_{i}$ is the ground truth, and $\hat{y_{i}}$ is the predicted output. The total loss is the weighted sum of the localization loss and classification loss.
(3)$$L\left(x,c,p,g\right)=\frac{1}{n}\left(L_{cls}\left(x,c\right)+\alpha L_{loc}\left(x,p,g\right)\right),$$
where *c* is class confidence; α = 1, which is similar to SSD [37], is used for fair comparison.

We also modified the cross-entropy and smoothL1 loss to handle the class imbalance problem. The proposed loss function considers the number of instances of each species in the training dataset and re-weights both the classification and localization loss to minimize the effect of the dominant class. The proposed class-aware classification and localization losses can be defined as:(4)$$\begin{array}{cc} L_{clsa} & =\frac{1-\left(\frac{n_{s}}{n}\right)}{1-{\left(\frac{n_{s}}{n}\right)}^{\eta }}L_{cls},\\ L_{loca} & =\frac{1-\left(\frac{n_{s}}{n}\right)}{1-{\left(\frac{n_{s}}{n}\right)}^{\eta }}L_{loc}, \end{array}$$
where $L_{clsa}$ is class-aware classification loss and $L_{loca}$ is class-aware localization loss; $n_{s}$ is the number of training instances per species, *n* is the total training samples, and η is a hyper-parameter. We use η = 4 for this training; $n_{s}<<n$. When $n_{s}$ is small and the value of η increases, the value of the multiplying term ($\frac{1-\left(\frac{n_{s}}{n}\right)}{1-{\left(\frac{n_{s}}{n}\right)}^{\eta }}$) increases, which maximizes the weight of the less represented class. The hyper-parameter values need to be tuned to get maximum performance. Generally, increasing the η value increases the value of the multiplying term of the loss functions, but it may not increase the overall performance due to its effect on other classes. Therefore, the best value of the hyper-parameter for maximum performance for a given dataset needs to be tuned starting with η = 1. The class balanced loss proposed by Cui et al. [42] improves the classification problems with class imbalance. Our class-aware loss function can work for classification and object detection with class imbalance datasets. The experimental result on the SEAMAPD21 dataset shows our loss gives a better result than the class balanced loss [42] (see Section 5 for details). The class-balanced (CB) loss for *y* class with $n_{y}$ training samples can be written as:(5)$$CB_{loss}=\frac{1-\beta }{1-\beta^{n_{y}}}L\left(x,y\right),$$
where *L*(*x*, *y*) can be cross-entropy loss or focal loss, and β is a hyper-parameter. We use β = 2 similar to the class balanced loss [42] for fair comparison, and this also gives best performance.

The authors considered the class-balanced loss for a classification problem, so they reweighted only the classification loss, such as focal loss. However, for other tasks, such as object detection, the localization loss is also affected by the class imbalance. Therefore, we consider the class-aware and class-balanced loss functions reweighting to both the localization and classification losses. We modified the class balanced loss for object detection as follows:(6)$$Objective_{loss}=\frac{1}{n}\left(\frac{1-\beta }{1-\beta^{n_{y}}}L\left(x,y\right)+\alpha \left(\frac{1-\beta }{1-\beta^{n_{y}}}\right)L_{loc}\left(x,p,g\right)\right),$$
where $n_{y}$ refers to the number of instances per species rather than the number of samples for each class in the class-balanced loss; α = 1 is used.

Different regularization techniques have been proposed to optimize network training [55,56,57]. In [55], a two-stage training method, pretraining and implicit regularization, was proposed to minimize the effect of overfitting. In the pretraining phase, the image representations were extracted. Next, in the implicit regularization stage, the feature boundary is regularized by retraining the network based on the results from the first stage. Training deep learning models, starting with a low learning rate and gradually increasing, becomes the common trend and improves the network’s performance [56,57]. In this work, the MultiStepLR scheduler is used to optimize the stochastic gradient descent (SGD) optimizer with an initial warm-up training and an initial learning rate of 10${}^{-3}$. The learning rate gradually decreases with a weight decay of 5 × 10${}^{-4}$. Additionally, a momentum of 0.9, gamma of 0.1, and batch size of 32 are used. The gamma value shows how much the learning rate drops at each step. All the experiments are done using Tesla v100 GPU running on Centos 7.8 operating system.

## 5. Results and Discussions

### 5.1. Quantitative Analysis

The commonly used evaluation metric in object detection is mean average precision (mAP), which we use to measure the model’s performance. The network predicts the average precision of each fish species, and the mAP is calculated for the overall class. Table 1 and Table 2 show the experimental result of our model on the SEAMAPD21 dataset and Pascal VOC public object detection dataset. The SEAMAPD21 dataset is challenging for the network to learn essential features due to low resolution and background and foreground similarity. Because of these challenges, the mAP of the network in the SEAMAPD21 fish dataset is less than that of the VOC dataset. The proposed class-aware loss function significantly improves the model’s performance, as shown in Table 1. When we randomly split the dataset into training, validation, and test data, some species might not have either validation or test data, especially those with fewer than ten images. Due to this issue in the dataset, we report only 82 species for the recognition results in this work. There is an increment of 14.42 mAP when the class-aware loss is used with MobileNetv3-large backbone, which is a 79.70% improvement over the original implementation. The model with a VGG16 backbone outperforms the MobileNetv3-large network even though the total parameters for VGG16 are almost 27 times greater. Increasing the resolution of the input image from 300 × 300 to 512 × 512 improves the model’s performance, as shown in Table 1 and Table 2. Therefore, choosing a smaller or larger backbone is a trade-off between high detection accuracy and low computational power. We also modified the class-balanced loss [42] for object detection (the class-balanced loss was proposed for a classification task), and we trained the model to compare it to our proposed loss function. The model’s mAP using the modified class-balanced loss with VGG512 backbone is 51.64%, which is lower than the mAP using the proposed class-aware loss of 52.75%.

Table 2 shows that the model outperforms the original SSD [37] network in 300 × 300 and 512 × 512 resolutions. The Pascal VOC dataset is relatively balanced, so it does not have a class imbalance issue like the SEAMAPD21 dataset. We also trained the model with the MobileNetv3-large backbone, which gave a comparable performance with a smaller number of parameters. When the model is trained with class-aware loss, the AP of many species increases; even those species with less than ten AP got more than 40. The AP of each species is shown in Appendix A
Table A1. Some fish species have zero AP, especially in the original cross-entropy loss implementation; however, their performance improves when the proposed loss is used (zero is a rounded value because the value is less than one). The AP of some fish species with more samples decreased when the proposed loss was used. However, the overall AP was improved.

Generally, the number of samples of each fish species, the resolution of each image, the similarity between some fish species, and the similarity of the foreground and background for some fish species contribute to the AP increment or decrement. As a rule of thumb, more samples of each species give better AP than a smaller number of samples if other factors are constant. Higher resolution also improves the detection performance; however, it increases the model parameters. Additionally, the network’s performance is reduced when the foreground is similar to the background.

### 5.2. Qualitative Analysis

We provide the qualitative outputs in Figure 5, Figure 6 and Figure 7 for VGG300, MobileNetv3, and VGG512 backbones, respectively. Using the VGG16 backbone at different image sizes, 300 × 300 and 512 × 512, gives an excellent qualitative output. The 300 × 300 image size almost gives as good a result as 512 × 512 except for some detections. For example, in row two, in the right image of Figure 5, there are six detections, but there are seven detections for VGG512 in Figure 7, which means one fish is not detected in Figure 5. However, the MobileNetv3 backbone has many missed detections compared to the VGG16 backbone. For example, for rows two and three, in the right images of Figure 6, there are five and one detections, but there are seven and four detections in Figure 7 for the corresponding images.

## 6. Conclusions

In this study, the fish species recognition task is formulated as an object detection task and evaluated on underwater images in the SEAMAPD21 fish species dataset. The MobileNetv3-large and VGG16 backbone networks are used as feature extractors. The experimental result on the SEAMAPD21 dataset shows that a model with large backbone networks can generalize better, but it increases the computational burden of the system. Although the VGG16 backbone network outperforms the MobileNetv3-large backbone network, it is 27 times larger, which increases the computational cost of the overall system, so it may not be suitable for real-time implementation. Therefore, choosing the best backbone network can be a trade-off between performance and processing speed. The SEAMAPD21 dataset is a highly imbalanced dataset, so the dominant class affects the network’s detection. The class-aware loss function is proposed to handle the class imbalance. The proposed loss function considers the instance of each species and reweights the loss function at each iteration to minimize the effect of the dominant class. The experimental result shows the effectiveness of the proposed loss function. The proposed loss function can be adapted to any object detection or classification task for an imbalanced dataset.

The existing backbone networks are used for this model. The backbone networks were designed for general-purpose computer vision tasks. To detect small objects, such as fish, and accurately extract features of challenging datasets, especially underwater images, specially designed feature extraction networks are essential. Because of the pooling operation in deep networks, there is information loss at each operation. Information loss affects small objects due to their fewer features at each layer. Due to these issues, there were missed detections in some cases, especially using the MobileNetv3 backbone. Additionally, we did not deploy the model in real-time to test the network’s performance. In future work, we will design a suitable backbone network to detect small objects and deploy the model in a low-power device for real-time implementation.

## Figures and Tables

**Figure 2 sensors-22-08268-f002:**
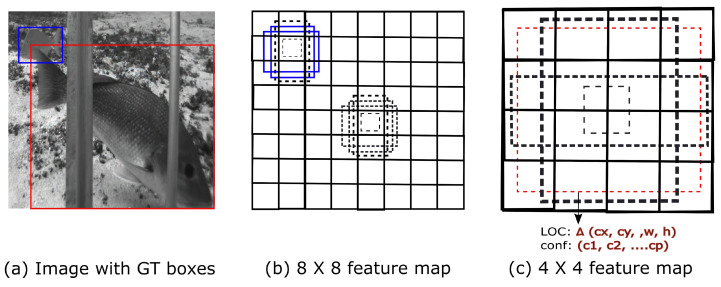
Overview of SSD detection [37]. (**a**) SSD takes an input image and ground truth boxes for each object during training. In a convolutional fashion, at each location, different small sets (e.g., 4) of default boxes of different aspect ratios in several feature maps with different scales (e.g., 8 × 8 and 4 × 4 in (**b**,**c**)) are evaluated. For each default box, the shape offsets and the confidences for all object categories (($c_{1},c_{2}$, …, $c_{p}$)) are predicted; $c_{x}$, $c_{y}$, *w*, and *h* refer to the *x* center, *y* center, width, and height of the bounding box, respectively.

**Figure 3 sensors-22-08268-f003:**
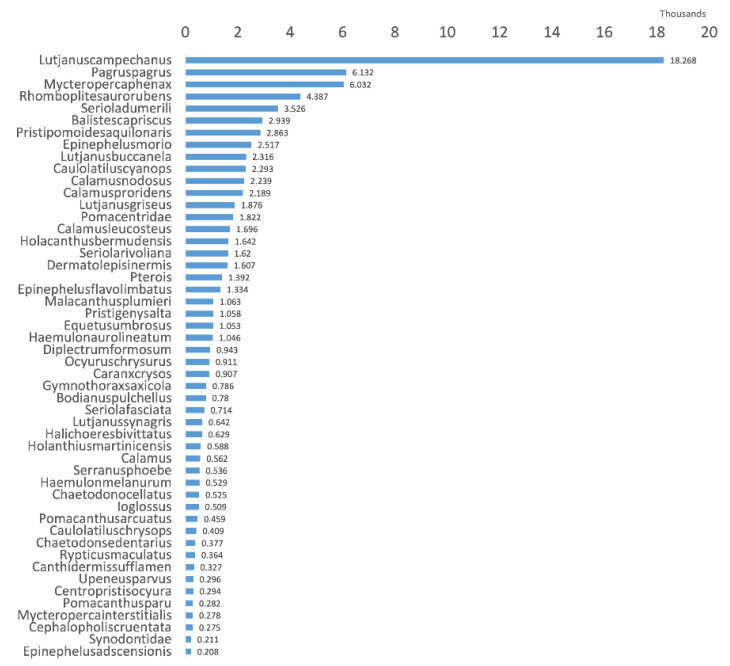
The sample number of occurrences per species distribution in SEAMAPD21 [7] shows a highly imbalanced structure.

**Figure 4 sensors-22-08268-f004:**
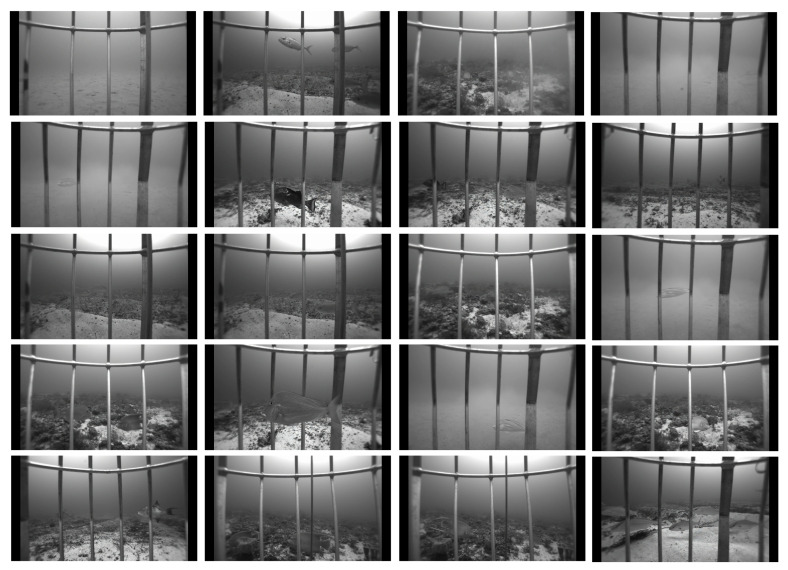
Sample images from the SEAMAPD21 dataset. The images are almost similar to the background, which makes the identification more challenging. Some of the fish images are challenging even for humans to detect. There might be occlusion due to vertical bars or other fish as well.

**Figure 5 sensors-22-08268-f005:**
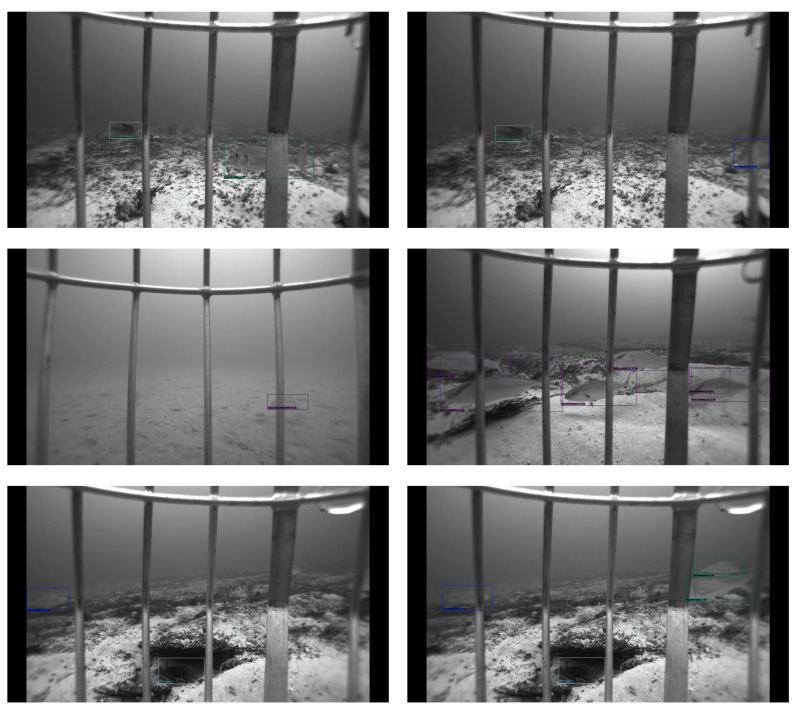
VGG300 backbone qualitative outputs. All fish species are detected in the sample images except the bottom left image. In the bottom left image, one fish is not detected, which is on the middle right side of the image. It is not easy to spot the missed fish, even for humans.

**Figure 6 sensors-22-08268-f006:**
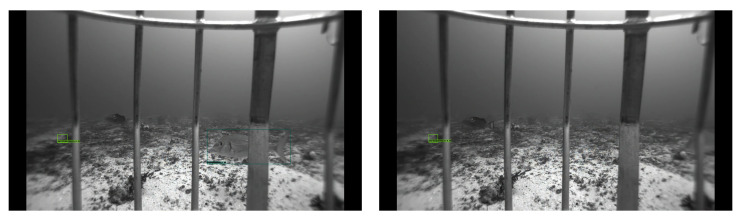
MobileNetv3 backbone qualitative outputs. These sample outputs show the missed detection using the MobileNetv3 backbone, whereas the VGG backbone detects them. One and two fish, respectively, are not detected in the first and second images of the first row. There is the same number of fish missed detection in the second row. However, two and three fish are not detected from the last row of images.

**Figure 7 sensors-22-08268-f007:**
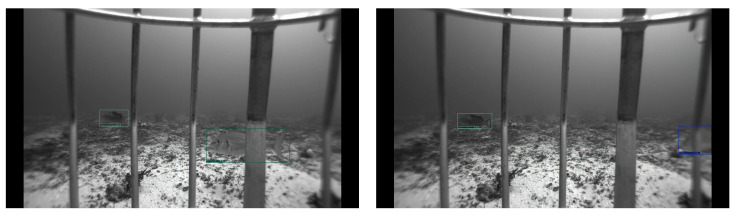
VGG512 backbone qualitative outputs. All fish species in each image are detected with high confidence.

**Table 1 sensors-22-08268-t001:** Experimental result (mAP %) of 82 species on the SEAMAPD21 dataset, where CE stands for cross-entropy loss and CACE stands for class-aware cross-entropy loss; ↑ means increment. VGG300 is the VGG16 backbone with an image size of 300 × 300, and VGG512 is for an image size of 512 × 512.

Backbone	CE Loss mAP	CACE Loss mAP	mAP ↑	Time (FPS)
MobileNetv3-large	18.09	32.51	14.42	105
VGG300	40.00	48.99	8.99	67
VGG512	42.79	52.75	9.96	54

**Table 2 sensors-22-08268-t002:** The experimental result (mAP %) comparison of the proposed model and the original SSD model on the Pascal VOC dataset. VGG300 is the VGG16 backbone with an image size of 300 × 300, and VGG512 is for an image size of 512 × 512.

Backbone	Original SSD	Proposed Model
VGG300	74.30	78.28
VGG512	76.8	80.61
MobileNetv3-large	-	70.79

## Data Availability

The dataset is available at https://github.com/SEFSC/SEAMAPD21 (accessed on 18 May 2022). The given link is updated every time. Therefore, it may have more datasets than we have used for this experiment.

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
