# Peer review of "Class-Aware Fish Species Recognition Using Deep Learning for an Imbalanced Dataset"

_sensors, 2022, doi:10.3390/s22218268_

Round 1

Reviewer 1 Report

The research topic is interesting.

(1) The reason for using mobilenetV3 should be given further. Its specific structure and advantages over other models should be described in detail. Can VGGNet be replaced by more advanced network models that introduce residual modules?

(2) More information about the experimental dataset is encouraged to be reported or presented as the environment appears to be very challenging.

(3) The difficulty brought by the integration of multiple models for the optimization of the objective function should be analyzed. Whether the convergence of the objective function will be affected.

(4) Since mobilenet is a lightweight neural network for edge devices, its inference speed should be reported.

(5) The key point of deep learning is whether it can be well generalized in the unknown environment. Therefore, the author should refer to and discuss the regularization and generalization of deep learning in the Related Work, such as “Improvement of generalization ability of deep CNN via implicit regularization in two-stage training process,” IEEE Access, vol. 6, pp. 15844-15869, 2018. “Cloud shape classification system based on multi-channel cnn and improved fdm.” IEEE Access 8 (2020): 44111-44124. “GPU-accelerated Faster Mean Shift with Euclidean distance metrics.” IEEE 46th Annual Computers, Software, and Applications Conference (COMPSAC). IEEE, 2022. B. Jin, L. Cruz and N. Gonçalves, "Pseudo RGB-Face Recognition," in IEEE Sensors Journal, 2022, doi: 10.1109/JSEN.2022.3197235.

Author Response

Response to Reviewer 1 Comments

Point 1: The reason for using mobilenetV3 should be given further. Its specific structure and advantages over other models should be described in detail. Can VGGNet be replaced by more advanced network models that introduce residual modules?

Response 1:  We want to thank you for your insightful comments on improving the manuscript.

We presented the reasons why the MobileNetv3 network was used for the experiment. We included additional details and rewrote it to clarify why the MobileNetv3 feature extractor was selected for the experiment. Other backbone networks, such as residual networks, can replace the VGG backbone. However, not all backbones give the same /consistent output for different applications. The VGG network is large and computationally intensive, unsuitable for real-time processing due to many parameters. The SSD model is designed with VGG as a backbone network. So, we kept the VGG backbone to make a fair comparison with the SSD network. We do not think the residual networks can improve the model's performance, but the model can be more lightweight with residual networks. From the experimental results of using VGG and from experience, we can say that VGG is an efficient network in terms of performance other than the computational burden.

Point 2: More information about the experimental dataset is encouraged to be reported or presented, as the environment appears to be very challenging.

Response 2: We included additional information about the dataset under the dataset and data augmentation subsection. A separate paper was also published previously. So, we are citing the paper for interested readers to check the details of the dataset.

Point 3: The difficulty brought by the integration of multiple models for the optimization of the objective function should be analyzed. Whether the convergence of the objective function will be affected.

Response 3: We included the optimization techniques and how the parameters are tuned during training in section 4.

Point 4: Since mobilenet is a lightweight neural network for edge devices, its inference speed should be reported.

Response 4:  We updated the manuscript with the inference speed (Table 1).

Point 5: The key point of deep learning is whether it can be well generalized in an unknown environment. Therefore, the author should refer to and discuss the regularization and generalization of deep learning in the Related Work, such as “Improvement of generalization ability of deep CNN via implicit regularization in two-stage training process,” IEEE Access, vol. 6, pp. 15844-15869, 2018. “Cloud shape classification system based on multi-channel cnn and improved fdm.” IEEE Access 8 (2020): 44111-44124. “GPU-accelerated Faster Mean Shift with Euclidean distance metrics.” IEEE 46th Annual Computers, Software, and Applications Conference (COMPSAC). IEEE, 2022. B. Jin, L. Cruz and N. Gonçalves, "Pseudo RGB-Face Recognition," in IEEE Sensors Journal, 2022, doi: 10.1109/JSEN.2022.3197235.

Response 5: Thank you for suggesting the papers. We included the suggested works to strengthen our manuscript. 

Reviewer 2 Report

In general, this paper is well organized in all sections, the paper provides a new methodology for Fish species recognition. Experimental results demonstrate the robustness of proposed method against others.

Author Response

Response to Reviewer 2 Comments

In general, this paper is well organized in all sections, the paper provides a new methodology for Fish species recognition. Experimental results demonstrate the robustness of proposed method against others.

Thank you for your insightful comment and for accepting the paper.

Reviewer 3 Report

Minor revision

This manuscript proposed a model which consists of MobileNetv3-large and VGG16 backbone networks and an SSD detection head. And a class-aware loss function is also proposed to solve the class imbalance problem of existing dataset.

Introduction

(1) Instead of introducing more basic research, this section should introduce the research progress related to this article.

(2) This section lists the contributions made in this article. Although these contributions are innovative, more contributions should be listed.

(3) Vision technology applications in various engineering fields, should also be introduced for a full glance of the scope of related areas. The first paragraph introducing the research topic may present a much broad and comprehensive view of the problems related to your topic with citations to authority references 

(Wu, F., Duan, J., Ai, P., Chen, Z., Yang, Z., & Zou, X. (2022). Rachis detection and three-dimensional localization of cut off point for vision-based banana robot. Computers and Electronics in Agriculture, 198, 107079.

 Tang, Y., Zhou, H., Wang, H., Zhang, Y. (2022).  Fruit detection and positioning technology for a Camellia oleifera C. Abel orchard based on improved YOLOv4-tiny model and binocular stereo vision, Expert Systems with Applications, 211:118573.)

The proposed Method and Implementation Details

(4) This section introduces the structure of the network proposed in this paper, in which the loss function is innovative.

Results and Discussions

(5) This section shows the performance comparison between the proposed network and the original network. The network in this paper has significant performance improvement, which is a good innovation.

(6) This section also qualitatively analyzes the shortcomings of the network proposed in this paper. Although the performance has improved to a certain extent, there are still missing targets, which needs to be optimized.

Author Response

Response to Reviewer 3 Comments

Point 1: Introduction

  1. Instead of introducing more basic research, this section should introduce the research progress related to this article.
  2. This section lists the contributions made in this article. Although these contributions are innovative, more contributions should be listed.
  3. Vision technology applications in various engineering fields, should also be introduced for a full glance of the scope of related areas. The first paragraph introducing the research topic may present a much broad and comprehensive view of the problems related to your topic with citations to authority references 

(Wu, F., Duan, J., Ai, P., Chen, Z., Yang, Z., & Zou, X. (2022). Rachis detection and three-dimensional localization of cut off point for vision-based banana robot. Computers and Electronics in Agriculture, 198, 107079.

 Tang, Y., Zhou, H., Wang, H., Zhang, Y. (2022).  Fruit detection and positioning technology for a Camellia oleifera C. Abel orchard based on improved YOLOv4-tiny model and binocular stereo vision, Expert Systems with Applications, 211:118573.)

Response Introduction: Thank you for your insightful comments to strengthen our manuscript. The related research on fish detection and classification is presented in the related section. The introduction section describes general fish detection, our work introduction, and some background information. Additional contribution is added to the manuscript. We listed down contributions that significantly added value to the manuscript. Although we did many tasks during the overall process, we do not consider them a contribution because they are common in most deep-learning works. In the related section, we listed only works related to fish detection. However, we included the suggested works in the introduction section as general computer vision-related work to strengthen our work.

Point 2: The proposed Method and Implementation Details

  1. This section introduces the structure of the network proposed in this paper, in which the loss function is innovative

Response proposed Method and Implementation Details: Thank you for accepting this section.

Point 3: Results and Discussions

  1. ) This section shows the performance comparison between the proposed network and the original network. The network in this paper has significant performance improvement, which is a good innovation.
  2. This section also qualitatively analyzes the shortcomings of the network proposed in this paper. Although the performance has improved to a certain extent, there are still missing targets, which needs to be optimized.

Response results and discussions: Thank you for your insightful comments. We included a paragraph about future work to further improve the model’s performance.

Reviewer 4 Report

The paper is about Class-aware Fish Species Recognition Using Deep Learning for an Imbalanced Dataset. The paper is nicely written and I have the following comments:

1. Please elaborate more about the motivation of this work. Why is this work significant?

2. What are the main limitations of related work?

3. Why R-CNN and YOLO are not being used as base-line models?

4. How the hyperparameters of the model were tuned? are there any optimization techniques?

5. How are the results compared to related work?

6. What are the main limitations of the proposed model?

7. Add a paragraph to describe future work

Author Response

Response to Reviewer 4 Comments

Point 1:  Please elaborate more about the motivation of this work. Why is this work significant?

Response 1: Section3, Deep learning for fishery, is the motivation section. This section shows why this work is significant to the marine industry, especially the fish industry. We updated Section 3 and elaborated on the motivation of this work.  

Point2: What are the main limitations of related work?

Response 2: Deep learning-based research in the fish industry is limited compared to other computer vision applications. Most publicly available fish datasets are simple, and each image comprises a single fish. These datasets are not representative of the real world. In the real system, many fishes exist in a single video or image. Therefore, most of the existing related works did a classification task. This work cannot be deployed in the long run for real-time implementation. Most of the fish dataset also does not contain many species like our dataset. We formulated the problem as an object detection task to handle multiple fish in a single image and multi-class classification. A class imbalance is a common issue related to fish species due to the abundance difference in a specific area. None of the related works address the issue.

Point 3: Why R-CNN and YOLO are not being used as base-line models?

Response 3: R-CNN generates 2,000 bounding boxes for each image using a selective search algorithm. The redundant generation of 2,000 bounding boxes from each image increases the network’s computational burden. The latest version of R-CNN is faster R-CNN, which solves the problem of R-CNN and fast R-CNN networks. However, Faster R-CNN is a two-stage detection network consisting of a region proposal network and a detection network. Generally, Faster R-CNN and its variants are slow due to the separate region proposal network. YOLO is a single-stage detection and one of the most decorated networks for object detection problems. YOLO is a faster network, but most of YOLO’s variant networks struggle to detect small objects.

On the other hand, SSD has four predictions of different scales and aspect ratios, which helps to detect an object of different scales. As we mentioned in the detection head subsection, that’s the reason for choosing the SSD detection head. Additionally, we have used SSD for other tasks. So, we are more familiar with SSD than YOLO. We have to note that the latest version of YOLO might solve the small object detection issues. 

Point 4: How the hyperparameters of the model were tuned? are there any optimization techniques?

Response 4: Thank you for catching this. We overlooked the hyperparameter tuning. The hyperparameters are tuned based on the suggestions given by “Accurate, Large Minibatch SGD: Training ImageNet in 1 Hour”. Training deep learning models, starting with a low learning rate and gradually increasing, becomes the common trend and improves the network’s performance. In this work, the MultiStepLR scheduler is used to optimize the Stochastic Gradient Descent (SGD) optimizer with an initial warm-up training and an initial learning rate of 10− 3. The learning rate gradually decreases with a weight decay of 5 × 10− 4. Additionally, a momentum of 0.9, gamma of 0.1, and batch size of 32 are used. We included in the manuscript the detail of hyperparameter tuning.

Point 5: How are the results compared to related work?

Response 5: The related works for fish recognition are classification works. The datasets used are also different, which makes the comparison unfair. Our fish species task is based on object detection models, which is more complex than most fish classification tasks. To show the model's performance, we compared the result with the original model of SSD (Table 2).

Point 6: What are the main limitations of the proposed model?

Response 6: The existing backbone networks are used for this model. The backbone networks were designed for general-purpose computer vision tasks. To detect small objects, such as fish, and accurately extract features of challenging datasets, especially underwater images, specially designed feature extraction networks are essential. Because of the pooling operation in deep networks, there is information loss at each convolution operation. Information loss primarily affects small objects due to their fewer features at each layer. Due to these issues, there were miss detections in some cases, especially using the MobileNetv3 backbone. Additionally, we did not deploy the model in real-time to test the network's performance. In future work, we will design a suitable backbone network to detect small objects and deploy the model in a low-power device for real-time implementation. 

Point 7: Add a paragraph to describe future work

Response 7: Thank you for the suggestion. We will design a more suitable backbone network to solve the abovementioned issues. Future work is included in the manuscript.

Round 2

Reviewer 1 Report

All the comments are well revised and thus this paper is recommended for publication.

Reviewer 4 Report

The authors addressed my comments and I happy to accept the paper. However, their reply should also be included in the revised version of the paper.

Good luck